# Hepcidin Protects Yellow Catfish (*Pelteobagrus fulvidraco*) against *Aeromonas veronii*-Induced Ascites Disease by Regulating Iron Metabolism

**DOI:** 10.3390/antibiotics10070848

**Published:** 2021-07-12

**Authors:** Manquan Fu, Rui Kuang, Weicheng Wang, Yunzhen Yu, Taoshan Ai, Xiaoling Liu, Jianguo Su, Gailing Yuan

**Affiliations:** 1Department of Aquatic Animal Medicine, College of Fisheries, Huazhong Agricultural University, Wuhan 430070, China; fmq220@webmail.hzau.edu.cn (M.F.); kr2015@webmail.hzau.edu.cn (R.K.); wwc@webmail.hzau.edu.cn (W.W.); liuxl@mail.hzau.edu.cn (X.L.); sujianguo@mail.hzau.edu.cn (J.S.); 2Engineering Research Center of Green Development for Conventional Aquatic Biological Industry in the Yangtze River Economic Belt, Ministry of Education, Wuhan 430070, China; 3Wuhan Chopper Fishery Bio-Tech Co., Ltd., Wuhan Academy of Agricultural Science, Wuhan 430207, China; scs@wuhanagri.com (Y.Y.); ats888@126.com (T.A.)

**Keywords:** yellow catfish (*Pelteobagrus fulvidraco*), hepcidin, berberine hydrochloride, *Aeromonas veronii*, iron metabolism

## Abstract

*Aeromonas veronii* (*A. veronii*) is one of the main pathogens causing bacterial diseases in aquaculture. Although previous studies have shown that hepcidin as an antimicrobial peptide can promote fish resistance to pathogenic bacterial infections, but the mechanisms remain unclear. Here, we expressed and purified recombinant yellow catfish (*Pelteobagrus fulvidraco*) hepcidin protein (rPfHep). rPfHep can up-regulate the expression of ferritin and enhance the antibacterial activity in primary hepatocytes of yellow catfish. We employed berberine hydrochloride (BBR) and Fursultiamine (FSL) as agonists and antagonists for hepcidin, respectively. The results indicated that agonist BBR can inhibit the proliferation of pathogenic bacteria, and the antagonist FSL shows the opposite effect. After gavage administration, rPfHep and the agonist BBR can enhance the accumulation of iron in liver, which may hinder the iron transport and limit the amount of iron available to pathogenic bacteria. Moreover, rPfHep and the agonist BBR can also reduce the mortality rate, bacterial load and histological lesions in yellow catfish infected with *A. veronii*. Therefore, hepcidin is an important mediator of iron metabolism, and it can be used as a candidate target for prevent bacterial infections in yellow catfish. Hepcidin and BBR have potential application value in preventing anti-bacterial infection.

## 1. Introduction

*Pelteobagrus fulvidraco* (*P. fulvidraco*) is an important freshwater economic fish in China [1]. Intensive farming and high-density stocking have caused many problems, such as bacterial diseases [2], fungal diseases, and parasitic diseases. *Aeromonas veronii* (*A. veronii*) is a conditional pathogen harmful to aquatic animals. It can cause abdominal enlargement and body surface ulcers in yellow catfish. It can even cause a large number of deaths. *A. veronii* produces a variety of virulence factors such as adhesion factors, hemolysin, enterotoxin, and cytotoxin [3,4]. It is also the main pathogen for *Ictalurus punctatus* [5,6], *Micropterus salmoides* [7], *Procambarus clarkia* [8], and rainbow trout [9]. The aquaculture industry mostly uses antimicrobial drugs such as antibiotics to control the occurrence and prevalence of diseases [10]. Excessive use of antimicrobial drugs has led to the increase of various antibiotic resistance genes (ARG) in *Aeromonas*, making the prevention and control more difficult [10,11].

The biological processes of all organisms require the participation of iron. Pathogens competitively seize the iron stored in the body during the invasion process, while the host isolates iron for nutritional immune defense [12,13]. Hepcidin is a main iron-regulating hormone, and it is induced by inflammatory factors in the early stage of infection [14,15].

Hepcidin is a small molecule, cysteine-rich peptide [14,16], and it was discovered in the early twentieth century in human. Hepcidin was first discovered in hybrid striped bass in 2002 [17]. Hepcidin is mainly secreted by the liver, and it enters into the blood [18]. Hepcidin binds to ferroportin, leading to its ubiquitination, degradation, and internalization [19], thus resulting in the decreased concentration of extracellular iron and the inhibited growth of ferrophilic bacteria. In recent years, the agonists and antagonists of hepcidin have become an important research field due to their high application value [20,21].

There is a close relationship between transferrin and the immune system. Transferrin binds to iron, creating an environment with iron of low levels, where few microorganisms can survive and prosper [22,23]. Transferrin exists in serum, and lactoferrin is secreted by the mucosa, as well as neutrophils at infection sites, and it serves as a host iron-withholding response, sequestering iron away from invading microorganisms. Apart from that, all cells contain ferritin, a protein composed of 24 subunits. Concerning ferritin, it can bind up to 4500 iron atoms, allowing it to be the major protein responsible for iron storage in cells [24]. Ferritin-bound iron, the major mechanism of iron storage in macrophages and liver hepatic cells [25], sequesters iron upon the increase of its intracellular levels, detoxifying and avoiding damage. Liver ferritin contains 50% of iron corporal reserves [26]. Iron homeostasis is controlled by a large group of iron-regulatory proteins.

Hepcidin can not only act as an immunomodulatory factor, and it can also directly kill bacteria by directly destroying the bacterial cell membrane and/or binding bacterial gDNA [19,27,28]. The hepcidin significantly increases the phagocytic activity and phagocytic index of phagocytes [29], and it may regulate the neutrophil recruitment in the immune response [30]. Our previous studies have also shown that the hepcidin peptides and recombinant proteins in grass carp play an important role in maintaining iron homeostasis and fighting against bacterial infections [31,32]. Hepcidin has been reported to significantly reduce the liver iron level and serum iron level in the fish [29,31]. However, there is relatively less research on the hepcidin in *P. fulvidraco*. Therefore, this study aims to explore the function of hepcidin protein and verify its antibacterial activity in vitro and in vivo and to identify protein substitutes by comparing the control functions between this protein and its substitutes. This study provides new information for the improvement of its resistance to ascites disease.

## 2. Results

### 2.1. Activity of Purified rPfHep

In order to obtain a high-purity fusion protein, 1 mmol/L IPTG was added into *E. coli* BL21 (DE3) plysS (Novagen) cells to induce protein expression overnight at 16 °C, and then affinity chromatography was performed. SDS-PAGE showed a target band at 35 kDa. The rPfHep protein band was relatively single (Figure 1A). The concentration of the purified protein was determined to be 1 mg/mL by the BCA method.

Western Blot (WB) assays of rPfHep protein and GST protein were performed, and the results showed that the bands were detected at 35 kDa and 26 kDa (Figure 1B), indicating that the rPfHep protein had high specificity, thus it could be used for follow-up experiments.

In order to investigate the function of rPfHep and to determine the best reaction time and concentration, hepatocytes were first incubated with 100 μM rPfHep for 6 h, and then the mRNA levels of ferroportin (FPN) and Ferritin were determined. After rPfHep treatment, the mRNA levels of FPN and Ferritin were significantly up-regulated (*p* < 0.001) (Figure 1C). In order to examine the change of ferritin protein in hepatocytes, the rPfHep at concentration of 100 μM and 200 μM was incubated with cells for 6 h, and the results indicated that ferritin level was significantly increased in a dose-dependent manner (*p* < 0.01; *p* < 0.05) (Figure 1D). No significant change in labile iron pool (LIP) was observed after 6 h rPfHep (100 μM) incubation, but LIP was significantly increased after 12 h incubation (*p* < 0.05), indicating LIP change slightly lagged behind (Figure 1E), compared with ferritin level. The above results indicated that rPfHep could regulate the iron content at mRNA level and protein level in hepatocytes.

### 2.2. Promotion of Bacterial Defense by rPfHep in P. fulvidraco Primary Hepatocytes

This study investigated the antibacterial protective effect of rPfHep on hepatocytes in vitro. The results showed that the cell survival and growth in the rPfHep group was better than those in the control group, the less cells were lysed and atrophied, whereas the cell survival rate in the rPfHep group was slightly lower than that in the uninfected group (Figure 2A). The plate counting method was used to count the bacteria in cells and culture medium. The results showed the number of bacteria in the rPfHep group was significantly smaller than that in the control group (*p* < 0.05), and no difference was observed between the GST protein group and the control group (Figure 2B). The qPCR results showed the mRNA levels of FPN, ferritin, and IL-6 in rPfHep group were significantly increased (Figure 2C–E). The significant up-regulation of IL-6 mRNA activated a strong antibacterial response. FPN has been reported to be internalized and degraded after being combined with rPfHep [19], but the mRNA level of FPN was up-regulated in this study, which was due to the fact that the actual detection time was shorter than the required reaction time. Since Ferritin and IL-6 are acute-phase proteins, their higher contents indicate a more severe inflammatory response to fight infection.

### 2.3. Identification of BBR and FSL as Hepcidin Agonists and Antagonists

In order to identify agonists and antagonists that can effectively act on the hepcidin gene of *P. fulvidraco*, we selected two agonists including adenine [33] and hydrochloric acid Berberine hydrochloride (BBR) [34] and the three antagonists including 17-β-estradiol (17-β) [35] and Guanosine 5′-diphosphate (GDP) [36], and fursultiamine (FSL) [37] according to previous reports. After incubating them with hepatocytes, the changes in mRNA levels of hepcidin were investigated. The results indicated that the incubation of Adenine and BBR (50 μM) with hepatocytes significantly increased hepcidin mRNA (*p* < 0.01) (Figure 3A), that GDP and 17-β had no significant effect on hepcidin mRNA in hepatocytes, and that the application of FSL significantly reduced hepcidin mRNA in hepatocytes (*p* < 0.05).

The above-mentioned results showed that the administration of rPfHep increased the antibacterial activity of cells. Identifying an easily accessible modulator that can directly regulate the hepcidin gene has become the current research direction. After the incubation of hepatocytes, respectively with BBR at the concentrations of 50 μM and 100 μM for 6 h, ferritin level was slightly increased with no statistical difference (*p* > 0.05). When hepatocytes were incubated, respectively with FSL at concentrations of 25 μM and 50 μM for 6 h, ferritin was significantly reduced (*p* < 0.05) (Figure 3B). When hepatocytes were incubated with BBR (100 μM) and FSL (50 μM) for 6 h, the labile iron pool (LIP) exhibited an opposite change trend between the BBR group and FSL group (*p* < 0.01) (Figure 3C). At the same time, the mRNA level of IL-6 in the BBR group was increased correspondingly (Figure 3D). Our results indicated that BBR and FSL had an effect on the proliferation of *A. veronii*. The number of bacteria in the BBR group was smaller than that in the control group, whereas the number of bacteria in the FSL group was the largest (*p* < 0.05) (Appendix A). In general, the effect of BBR on hepatocytes was similar to that of rPfHep. BBR restricted the proliferation of bacteria in the hepatocytes and exhibited a certain protective effect on the hepatocytes, whereas the effect of FSL exhibited an opposite effect on hepatocytes. 

### 2.4. Reduction in Mortality and Tissue Bacterial Load by rPfHep and BBR in Yellow Catfish

The relative percentage survival (RPS) was used to evaluate the protective effects of rPfHep and its agonist BBR on yellow catfish (*P. fulvidraco*) infected with *A. veronii*. *A. veronii* was injected intraperitoneally into the samples after two intragastric administrations. The early protective effects of rPfHep and BBR were extremely significant. Seven days after the infection, the mortality rate in the control group was 91.55%, and that in the rPfHep group was 76.06%, whereas that in the BBR group was 40.3%, and the RPS of rPfHep and BBR were 16.91% and 55.98%, respectively (Figure 4A). We applied Kaplan–Meier test to the analysis on mortality among groups. It was identified that the mortality rate in the rPfHEP group and BBR group was significantly lower than that of the control group (*p* < 0.01).

The number of bacteria in a unit tissue can reflect the health of the fish. Three days after *A. veronii* infection, the bacterial load in the gills, liver, intestine, kidney, and spleen of the fish was measured. The results showed that the bacterial load of the tissues in all the treatment groups was lower than that in the control group, which was consistent with the investigation results of mortality rate (Figure 4B–F).

### 2.5. Alleviation of A. veronii-Induced Inflammation by rPfHep and BBR

The gill filaments of healthy yellow catfish are neatly distributed, and the gill pieces are arranged in an orderly manner. However, in this study, the proliferation of gill filament epithelial cells was observed in the infected control group, and the gill pieces were severely proliferated and fused into a plate shape. The lesions in the rPfHep group were alleviated, compared with those in the infected control group. In rPfHep treatment group, the gill fragments were occasionally swollen but they did not fuse into a plate shape, and the thickness of the epithelial cell layer was thinner than that in the control group. The pathological changes in the BBR group were the slightest, the gill pieces were neatly arranged without shedding and necrosis (Figure 5A). The outline of healthy yellow catfish liver cells was clear, and the nucleus was located in the center of the cell. In the infected control group, hepatocytes were enlarged, and vacuoles were observed. In infected control group, hepatocyte nuclei were pyknotic and shifted from the cell center; hepatocyte cords were arranged disorderly; hepatic sinusoids were congested. Lower degrees of hepatocyte vacuolation and hepatic sinusoidal congestion were observed in rPfHep treatment group than in infected control group. The BBR treatment group exhibited the mildest lesions with occasional hepatocellular swelling (Figure 5B).

After a short-term exposure to rPfHep and BBR for one day, the total protein (TP) and superoxide dismutase (SOD) levels in the rPfHep and BBR treatment group were not significantly different from those in the control group, but the lysozyme (LZM) level in treatment group was increased. Three days after infection, the SOD in the rPfHep treatment group was higher than that in the other groups (*p* < 0.01), and the TP and LZM in the BBR group were significantly higher than those in the other groups (*p* < 0.01) (Figure 5C–E). Further, the activities of LZM, SOD and TP in the serum were measured, and the results indicated that these activities in the rPfHep group and the BBR group after infection were higher than those of the control group, which was conducive to the elimination of bacteria and the return to body steady state.

### 2.6. Regulation of Individual Iron Levels by rPfHep and BBR

Under normal conditions, the ferritin of macrophages and hepatocytes will be stained into spot-like blue particles via Perls’ Prussian blue staining [38]. Before infection, the rPfHep group had many large blue granules, indicating that rPfHep treatment can increase liver iron levels. After infection, the blue particles in each group were increased, indicating that the infection induced the transfer of iron ions, and the increase in blue particles in the BBR group was the most obvious (Figure 6A). As shown in Figure 6B, the acute-phase ferritin protein in the serum was significantly increased after infection (*p* < 0.01) (Figure 6B). The serum iron level in the rPfHEP group and BBR group was significantly lower than that of the control group (*p* < 0.01) (Figure 6C).

## 3. Discussion

Yellow catfish had only one hepcidin gene, which is similar human [39]. The human hepcidin gene also functions as an antimicrobial peptide and maintains iron metabolism in the body [40,41]. Rats and mice have been found to have two hepcidin genes, and these two genes had similar structures, but performed different functions. Fish are lower vertebrates, and some fish contain a variety of hepcidins. For example, there are six types of hepcidins in large yellow croaker [42], and two types of hepcidins in perch [43]. The difference in hepcidin gene number indicates the difference in evolution and function. There is already a lot of evidence that marine animals are the ancestors of land mammals. In the process of species evolution, survival demands also vary with changes in the environment. The study of lower vertebrates helps to analyze the evolutionary history of animals and the process of species gene dominance [43,44]. The study of the hepcidin function of yellow catfish may be of positive significance to the study of animal evolutionary history.

Hepcidin gene (*Trachidermus fasciatus*) expressed by *Pichia pastoris* has a pattern-recognition function and agglutination activity against a variety of bacteria [45]. The SA-hepcidin1 gene of *Scatophagus argus* is distributed in various tissues and has antibacterial effect, and the SA-hepcidin2 is only distributed in the liver and has specific antiviral activity [46]. This study only investigated the regulation effect of hepcidin on iron level in yellow catfish, and further studies are needed to investigate more functions of this gene. The antibacterial mechanism of antimicrobial peptides is usually to dissolve the bacterial cell membrane, allow the content to flow out, and aggregate bacteria through the strong positive charge characteristics [47,48]. Certain antimicrobial peptides need to form a specific structure to sterilize at a specific pH value [49]. Studies have shown that the mature peptide of synthetic hepcidin exerts antibacterial functions against bacteria and fungi [29,45] by destroying the integrity of microbial membranes [42]. Therefore, the rPfHep expressed by the *E. coli* expression system may not be folded correctly during the production process, thus losing its direct bactericidal activity, which suggested that *Pichia pastoris* expression system or synthetic peptides may be a better choice for investigating hepcidin function.

The liver is the largest organ in the fish body, which is responsible for digestion, metabolism, and detoxification and it has complex immune regulation functions [23,50]. Since hepatocytes are experimentmal model, there have been extensive nutritional and toxicological studies on them. Liver cells are also involved in pathological processes, such as inflammation and immune disorders. The liver is the main organ secreting hepcidin [14], and hepcidin enters the intestine with the blood system to function [51]. Therefore, hepatocytes are a suitable material for studying the function of hepcidin. Since yellow catfish had no liver cell line, we isolated and cultured the primary hepatocytes of yellow catfish to examine the effects of hepcidin and its regulatory functions. This study investigated the effects of agonists and antagonists of rPfHep protein and hepcidin oncellular immune responses in vitro with cells as research materials.

Chelating iron or restricting bacterial access to iron is the host’s first defense line against pathogens [13,52]. Chilean salmon and Coho salmon have differences in expressing heme degradation and iron transport-related genes such as hepcidin, transferin, and haptoglobin, which in turn leads to differences in sensitivity to sea lice [53,54]. These confirmed that iron regulation might be an important mechanism for immune function during host infection. This immune mechanism is used in vertebrates to prevent or reduce bacterium, parasite, and blood parasite infections [53]. The regulation of iron is related to the key factors of fish immune response. The chain effect of hepcidin, ferroportin, lactoferrin, ferritin, and other factors can directly affect the availability of iron during pathogen infection [23,55,56], and it plays an important role in combating intracellular pathogens. This study detected the changes in serum ferritin protein and serum iron. Our results indicated that bacterial infection increased the level of ferritin and decreased serum iron. The application of rPfHep and hepcidin as well as its gene agonists enhanced the body’s antibacterial response and promoted the accumulation of iron in the liver, thus slowing down the development of infection [57,58].

BBR (Berberine hydrochloride) is a kind of light yellow isoquinoline alkaloid extracted from berberine-containing plants [59]. BBR is often used as bacteria to treat bacterium infection-induced gastrointestinal diseases such as diarrhea [60,61]. BBR alone or in combination with enrofloxacin has bactericidal effects on six common fish pathogens such as *Aeromonas hydrophila*, *Vibrio vulnificus*, and *Pseudomonas fluorescens* [62]. One previous study found that BBR could regulate iron metabolism in mice. After mouse liver cancer cell Hepa1-6 was exposed to BBR, the Smad1/5/8 pathway can be activated to promote the expression of hepcidin [34].

The BBR group exhibited an effective reduction in the mortality rate in yellow catfish, and a significant increase in the relative immune protection rate, compared with the rPfHep group. It should be noted that rPfHep displayed a good protective effect in the early stage of the experiment, but the effect in the later stage of the experiment is less good, which may be caused by the instability of the protein. Therefore, a more suitable drug administration strategy needs to be explored in the future. Our data indicated that BBR could be used for up-regulating hepcidin, thus it plays an active role in fish farming. In addition, our data indicated that BBR has the multiple advantages such as easy availability, low price, no obvious adverse reactions, or no bacterial drug-resistance, therefore it is a relatively desirable feed additive.

## 4. Materials and Methods

### 4.1. Experimental Animals

*P. fulvidraco* (50 ± 5 g) was obtained from the JiangXia Fish Breeding Base (Wuhan, China). In the fish farming tanks, a flow-through aquaculture system was utilized. In this system, the water quality in each tank was independent of the others. The fish were treated in accordance with the “Guidelines for the Care and Use of Experimental Animals”, which was approved by the Animal Ethics Committee of Huazhong Agricultural University.

In cell treatment experiment, the fish were fed with commercial feed daily (1.5% of fish body weigh). The experimental fish were starved for 24 h before liver extraction.

In individual experiment, the fish were divided into three groups: control group, rPfHep group, and Berberine hydrochloride (BBR) group with 69 fish per group and 23 fish in each tank from each experimental group except sampled fish. Short-term exposure was performed by gavage administration [63]. The rPfHep group was treated with 2 μg/g pGEX-KG-hepcidin fusion protein for two days [31,32]. The BBR group was treated with 300 mg/kg BBR dissolved in PBS [63,64]. The control group was treated with equal amount of 100 μL 0.65% sodium chloride solution (NaCl). Each group was added with 3/1000 glucose as protective agent. In order to investigate the effect of drugs on the iron level, liver iron level and serum iron level of livers and blood samples from 4 fish were detected after gavage administration. After 2-day continuous gavage administration, 300 μL *A.veronii* at the concentration of 10^7^ CFU/mL was injected intraperitoneally [47]. The mortality within seven days was observed. Three days after bacterial infection, blood was taken from the tail of 4 fish with a syringe. Iron level, enzyme activity, and ferritin parameters in the serum were determined. Liver and gill tissues were sampled and stored in 4% paraformaldehyde to compare the pathological changes and liver iron level. The 0.01 g of gill, liver, intestine, kidney, and spleen were stored in 1 mL PBS at 4 °C for further study.

### 4.2. Hepatocyte Isolation

Before sampling, fish were starved for 24 h. Hepatocytes were isolated from eight yellow catfish according to the previously reported method [63] with slight modification. Firstly, blood was removed from yellow catfish by cutting off the branchial arch, and disinfected with 75% alcohol. After blood removal, the liver was carefully excised from the abdominal cavity, placed onto a plastic Petri dish, and rinsed twice with phosphate-buffered saline (PBS, pH 7.4, 4 °C) containing amphotericin-B (25 μg/mL), streptomycin (100 μg/mL), and penicillin (100 IU/mL). Then, the liver was aseptically minced into 1 mm^3^ pieces with scalpel and scissors, and the tissue was digested by 0.25% sterile trypsin at room temperature on a shaker for 30 min, and the digestion was terminated with M199 medium containing 10% FBS every 5 min. The cell suspension was collected. Then, the isolated hepatocytes were purified through nylon sieve with 200 μm mesh size. Hepatocytes were put into a 15 mL sterile centrifuge tube and added with RCLB (red blood cell lysis buffer) at a ratio of 1:4 to remove excess red blood cells (RBC). Hepatocytes were centrifuged at low speed (1000 rpm/min for 5 min) and washed twice with PBS to remove debris. Finally, the purified hepatocytes were resuspended in M199 medium containing 1 mmol/l-glutamine, 5% (*v*/*v*) FBS, penicillin (100 IU/mL), and streptomycin (100 μg/mL). Hepatocytes were counted using a hemocytometer based on the Trypan blue exclusion method. Hepatocytes with more than 95% cell viability were used for the subsequent experiments. The hepatocyte cell suspension (CS) at the cencentration of 1 × 10^6^ cells per mL was plated into 25 cm^2^ flasks.

### 4.3. Bacterial Activity

*A. veronii* used in the experiment was isolated from yellow catfish suffering from ascites in 2018 by Yangtze River Fisheries Research Institute, Chinese Academy of Fishery Sciences. The bacteria was named XJH01 and was donated to our laboratory to expand the species preservation [65].

The strain XJH01 was stored in the −80 °C cryogenic refrigerator, and cultured in the BHI liquid medium at 30 °C in a shaker (180 rpm/min). After the bacteria solution OD600 reached 0.5, the strain was collected through centrifugation at 5000 rpm/min for 10 min. The supernatant was discarded. The precipitates were resuspended in sterile PBS, centrifugated, and washed for three times.

An appropriate amount of sterile PBS was added to the precipitates, and the concentration of the bacterial solution was determined by the red blood cell counting and spectrophotometer counting.

According to the number of cultured primary hepatocytes, the bacteria were diluted with sterile PBS to multiplicity of infection (MOI) of 10 for cell treatment experiment. The bacteria were diluted with sterile PBS to 1 × 10^6^, 1 × 10^7^, 1 × 10^8^, and 1 × 10^9^ CFU/mL for individual experiment.

### 4.4. Expression and Purification of Recombinant Hepcidin

DNAs encoding the open reading frame (ORF) of hepcidin (GenBank: EU257703.1) were inserted into the *E. coli* expression vector pGEX-KG (Novagen) using the primers in Table 1. The recombinant construct (pGEX-KG-hepcidin, rPfHep) with the the restriction enzyme sites *Bam*H 1 and *Hin*d III and plasmid pGEX-KG were transformed into *E. coli* BL21 (DE3) plysS (Novagen) cells. The transformed cells were induced with 0.1 mM Isopropyl β-d-thiogalactoside (IPTG) at 16 °C overnight and purified by the glutathione-S-transferase (GST) fusion system. Protein concentration was determined using a BCA protein assay kit (Beyotime, China) [31].

The bacterial cells were collected by centrifugation at 5000× *g* for 10 min, re-suspended in 20 mM PBS (pH 6.3), and high-pressure crushed on ice. The cell debris was removed by centrifugation at 12,000× *g* for 1 h, and the supernatant containing proteins passed through glutathione beads (Smart-Lifesciences, Jiangsu, China). The purified recombinant protein rPfHep and purified protein GST were then detected by SDS-PAGE [32,66].

### 4.5. Western Blot Analysis

Western blot analysis was used to confirm the existence and molecular weights of the obtained rPfHep protein. The recombinant PfHep protein and empty vector (pGEX-KG)-expressed GST-tagged protein was separated on a 12% SDS-PAGE gel, transferred to nitrocellulose filter membrane at 9 V for 30 min. The membrane was blocked with 5% skimmed milk at room temperature for 1 h, incubated with a 1:4000 dilution of a anti-GST tag mouse monoclonal antibody for 1 h, washed with 1 × TBS + 0.1% Tween and incubated with a 1:5000 dilution of HRP Goat Anti-Mouse secondary antibody (1:5000; 1 mg/mL) (ab6789, Abcam) for 1 h.

### 4.6. Intracellular Iron Measurement

Intracellular iron levels were evaluated using the previously reported fluorometric assay [67] with some modifications [29,68]. The calcein-acetomethoxy (CA-AM) was replaced with green fluorescent heavy metal indicator Phen Green FL (PG-FL; Invitrogen). Briefly, cells were washed twice in the PBH buffer (20 mM HEPES, 1mg/mL BSA in PBS) and incubated with PBH buffer containing 30 μM Phen Green FL (PG-FL; Invitrogen) at 28 °C for 30 min. PBH buffer was used to remove residue agents, and cells were washed twice with PBS. Subsequently, the cells were lysed with 0.2× PBS, and the lysates were incubated on ice for 30 min. Cell lysates were then centrifuged at 12,000× *g* for 10 min at 4 °C to remove cellular debris. PG-FL fluorescence intensity (F0) was measured at 490 nm (excitation)/528 nm (emission) with the Multiscan Spectrum microplate reader (SpectraMax i3x, Molecular Devices, Sunnyvale, CA, USA). After this initial measurement, 2, 2 -bipyridine (Sigma, Saint Louis, MO, USA) (10 μM) was added to strip iron from chelated PG-FL, and the fluorescence intensity (F1) was measured again using the same filtering combination. The change in fluorescence intensity was used to evaluate the intracellular labile iron pool (LIP).

### 4.7. RNA Extraction and qPCR Analysis

Total RNA was extracted from liver with RNAiso Plus (Takara, Dalian, China) according to the manufacturer’s instructions, followed by DNase I treatment. Total RNA was quantified by absorbance ratio of 260 nm to 280 nm with NanoDrop 2000 spectrophotometer (Thermo Scientific, Waltham, USA), and the total RNA quality was assessed by 1% agarose gel electrophoresis. The cDNA was synthesized using the HiScript^®^ II Q Select RT SuperMix (Vazyme, Nanjing, China) according to the manufacturer’s instruction for subsequent qPCR. All the cDNA products were diluted to 250 ng/μL, and the qPCR was performed using the AceQ^®^ qPCR SYBR^®^ Green Master Mix (Vazyme, Nanjing, China) on a Roche LightCycle^®^ 480 System (Roche, Basel, Switzerland) according to the manufacturer’s instructions. Iron-related genes including hepcidin, ferroportin, and ferritin and immune gene IL-6 were determined. All the primers used were shown in Table 1. A melting curve was constructed for every qPCR product to confirm the specificity of the assays. A dilution series were prepared to check the efficiency of the reaction. β-actin was used as the housekeeping gene. The 2^−ΔΔCT^ method based on the cycle threshold (CT) values was used to analyze gene expression levels.

### 4.8. Tissue Staining

The livers and gills were dissected and fixed immediately in 10% neutral buffered formalin for 24 h, dehydrated, paraffin-embedded, and sectioned. Sectioned samples (4 m) were mounted on aminopropyl-triethoxysilane-coated slides. Following the deparaffinization in xylene, sections were rehydrated, stained with Hematoxylin and eosin (HE) and prussian blue staining, and mounted with neutral gum, in which the images were captured.

### 4.9. Statistical Analysis

The qRT-PCR data were expressed as mean ± Standard Error of Mean (SEM) to reveal statistical differences between groups. The plots were drawn by the Graphpad Prism 5 software (San Diego, CA, USA). The experiments were carried out in triplicate. Statistical significance was expressed at three levels of *p*  <  0.05, *p*  <  0.01 and *p*  <  0.001.

## 5. Conclusions

This study reveals that hepcidin can be used to effectively resist bacterial infections through iron regulation. Berberine hydrochloride (BBR) and Fursultiamine (FSL) can act as effective agonists and antagonists of yellow catfish hepcidin. The rPfHep and BBR limit the available iron in liver cells, thus reducing the proliferation of *A. veronii* in cells. Short-term exposure to rPfHep and BBR can improve the survival rate of yellow catfish, reduce the bacterial load in the tissue, and alleviate tissue inflammation. The rPfHep and BBR can be used for fighting against infection by increasing the ferritin protein level, reducing serum iron level, and stocking iron in the liver. Due to its availability, stability, and high efficiency, BBR has a great potential to be used as an alternative medicine for the treatment of yellow catfish ascites disease in the future.

## Figures and Tables

**Figure 1 antibiotics-10-00848-f001:**
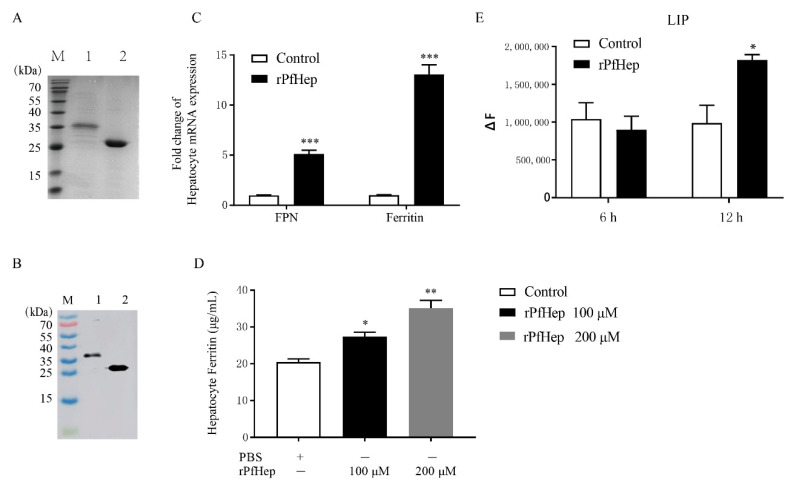
rPfHep is a protein regulating iron level. (**A**) SDS-PAGE analysis of purified pGEX-KG-hepcidin. M, markers; lane 1, pGEX-KG-hepcidin, lane 2, GST protein. (**B**) Western blotting assay of anti-GST antibody. M, markers; lane 1, pGEX-KG-hepcidin fusion protein with GST; lane 2, GST protein. (**C**) qRT-PCR assay of mRNA level of Ferritin and FPN (Ferroportin) in hepatocytes treated with rPfHep. (**D**) ELISA of Ferritin level in rPfHep-treated hepatocytes (100 μM and 200 μM). (**E**) Fluorescence assay of LIP level in hepatocytes treated with rPfHep. * *p* < 0.05; ** *p* < 0.01; *** *p* < 0.001 versus control.

**Figure 2 antibiotics-10-00848-f002:**
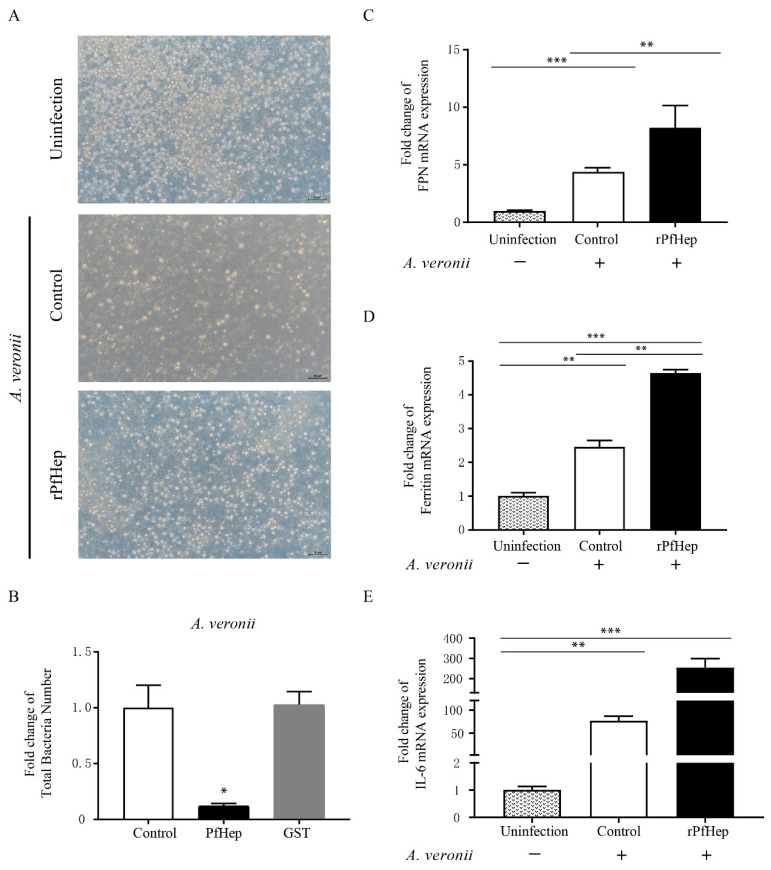
rPfHep activates a stronger and more effective antibacterial response of liver cells after. rPfHep-incubated (6 h) hepatocytes was stimulated by *A. veronii* for 3 h. (**A**) Comparison of hepatocytes activity among rPfHep group, Control group, and uninfection group. 100×. (**B**) Fold change of bacterial number of Control, rPfHep, and GST group. (**C**–**E**) qRT-PCR assay of the mRNA fold changes of FPN, Ferritin and interleukin-6 (IL-6). The data are expressed as mean ± SEM (*n* = 4). * *p* < 0.05; ** *p* < 0.01; *** *p* < 0.001 versus control.

**Figure 3 antibiotics-10-00848-f003:**
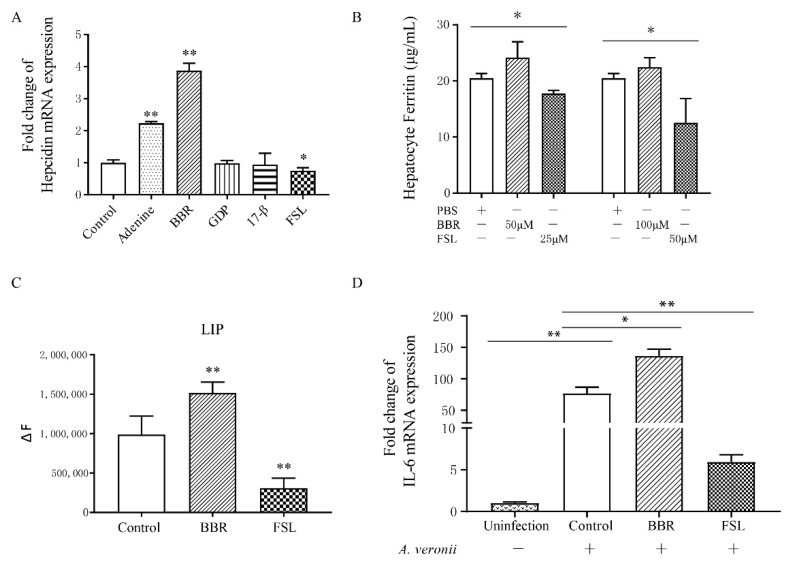
BBR and FSL regulate hepcidin as agonists and antagonists in *P. fulvidraco* hepatocytes. (**A**) Effects of Adenine, BBR, GDP, 17-β, and FSL on the mRNA level of hepcidin in hepatocytes. (**B**) ELISA of the ferritin protein level of after incubation of hepatocytes with different concentrations of BBR and FSL. (**C**) Fluorescence detection of LIP level after incubation of hepatocytes with BBR and FSL. (**D**) qRT-PCR assay of mRNA level of IL-6 after 6 h incubation of BBR or FSL with hepatocytes followed by 3 h infection with *A. veronii*. * *p* < 0.05; ** *p* < 0.01 versus control.

**Figure 4 antibiotics-10-00848-f004:**
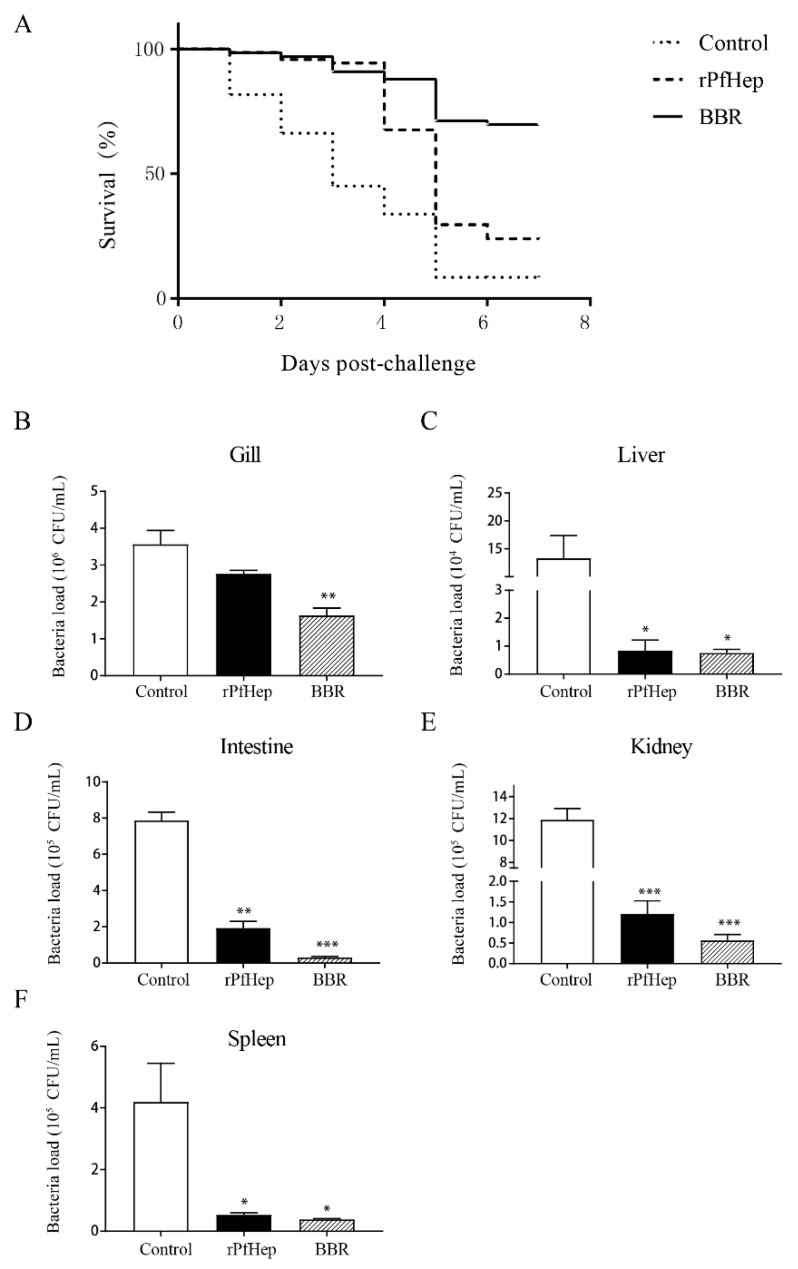
Effect of rPfHep and BBR on survival rate and tissue load of *P. fulvidraco*. (**A**) Effect of rPfHep and BBR on the survival rate of *P. fulvidraco* infected with *A. veronii*, and the mortality rate was recorded for 7 days after *A. veronii* infection in each group. (**B**–**F**) Bacterial load in gill, liver, intestines, kidney, and spleen tissues in *P. fulvidraco* three days after *A. veronii* infection with 0.01 g tissue added to 1 mL PBS. * *p* < 0.05; ** *p* < 0.01; *** *p* < 0.001 versus control.

**Figure 5 antibiotics-10-00848-f005:**
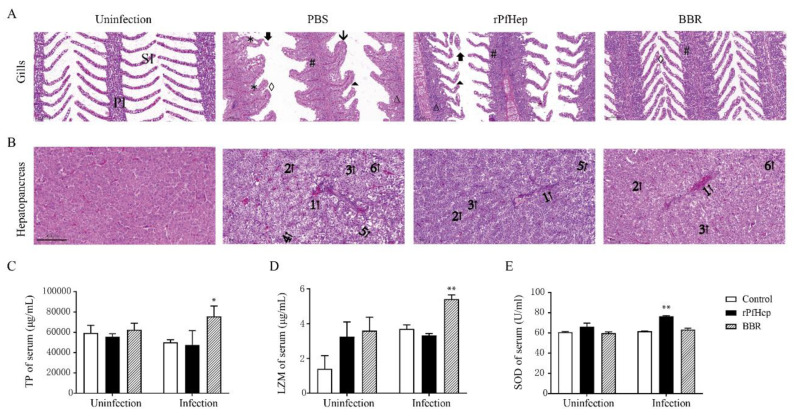
Effects of rPfHep and BBR on histopathological changes and serum biochemical indexes of *P. fulvidraco*. (**A**) The pathological sections of gills, Uninfection group, Control (infection) group, rPfHep group, BBR group, Gill filament (PI); Gill lamella (SI). In the control group, the proliferation of gill filament epithelial cells was observed, and the gill pieces were severely proliferated and fused into a plate shape. As a result, the histopathological changes of rPfHep group and BBR group were better than that of the control group. Epithelial hyperplasia (#); inflammatory cell infiltration (Δ); Curved gill lamella (⏶); Epithelial edema (◊); Epithelial cells separate from capillaries to form cavities (*); Shedding and necrosis of gill pieces (🡇); Gill lamella are severely proliferated and merge with adjacent gill pieces into a plate shape (🡳), stained with HE. (**B**) The pathological sections of liver. In the control group, hepatocytes were enlarged; inflammatory cell infiltration and even focal necrosis appeared in the liver. However, it was not so serious in the rPfHep group and BBR group. Hepatic sinusoidal congestion (1↑); Hepatocyte pyknosis (2↑); Liver cells are enlarged and vacuoles appear (3↑); Hepatocyte nucleus lysis gradually disappears (4↑); Focal necrosis (5↑); Inflammatory cell infiltration (6↑), stained with HE. (**C**–**E**) Total protein (TP), lysozyme (LZM) and superoxide dismutase (SOD) in serum of yellow catfish. * *p* < 0.05; ** *p* < 0.01 versus control.

**Figure 6 antibiotics-10-00848-f006:**
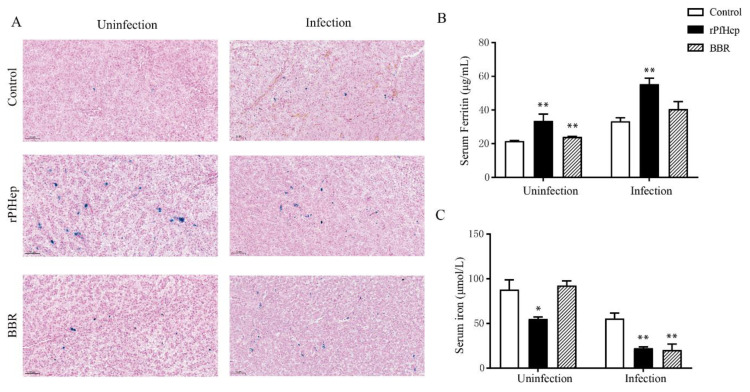
Regulation of liver iron and serum iron levels by rPfHep and BBR. (**A**) The left side is the liver section before infection, and the right side is the liver section after infection, perls’ prussian blue staining. (**B**) ELISA of serum ferritin protein level of yellow catfish. (**C**) Detection of serum iron level of yellow catfish with Nanjing built serum iron test kit. * *p* < 0.05; ** *p* < 0.01 versus control.

**Table 1 antibiotics-10-00848-t001:** Primer sequences used for PCR.

Gene	Primer	Primer Sequence (5′-3′)
PfHep	Forward	CGC*GGATCC*GCAGTACCTTTCTCTCAGAATG
	Reverse	CCC*AAGCTT*TTAGAACCTGCAGCAGAACC
Ferroportin (FPN)	Forward	AAAACGCTCGGCTCAAAGTG
	Reverse	GTAGCAAAACGTCAGCAGCC
Hepcidin	Forward	GCAGTACCTTTCTCTCAGAATG
	Reverse	TTAGAACCTGCAGCAGAACC
Ferritin	Forward	TGTCAAACGGCACTTCCTCA
	Reverse	TGTCATTGGTGCATCCCACTT
IL-6	Forward	CTCCAGACCAGAAGTGGGTTGA
	Reverse	CCCTTATAGGCGTAAATAGTCGTGTT
β-actin	Forward	TTCGCTTGGAGATGATGCT
	Reverse	CGTGCTCAATGGGGTACT

The underline incates the protected base, and italical part represents restriction site.

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
