# Peer review of "Hepcidin Protects Yellow Catfish (Pelteobagrus fulvidraco) against Aeromonas veronii-Induced Ascites Disease by Regulating Iron Metabolism"

_antibiotics, 2021, doi:10.3390/antibiotics10070848_

Round 1

Reviewer 1 Report

The authors tested whether hepcidin protects yellow catfish from A. veronii infections. This paper is interesting overall, but has some problems in experimental design and broad conclusions that the data do not fully support. 

  1. For fig 1B, the authors state that the western blot indicates that the rPfHep protein has high specificity. A western blot shows whether the antibody is specific to a protein, not the other way. Why do they see two bands for the GST control in lane 2?
  2. From fig 3A, the authors state that there was a significant decrease in hepcidin mRNA with 17-B and FSL. The fold change is minimal. I wouldn't consider that a significant change.
  3. I didn't understand what the first panel in Fig 3D is. I see no changes in any of those spots.
  4. In the experiments described in figure 4, the authors introduce a purified hepcidin through the gavage. Can they show the protein is stable under these conditions, and that it reaches the hepatocytes in an active form? If non, the authors can not make any conclusions from these data.  

Author Response

We would like to thank you for your kind help in reviewing our manuscript. We appreciate all the valuable and constructive comments and suggestions. They helped us improve our paper. To better present the manuscript, we have studied the comments scrupulously and have made explanations and corrections. Here is a summary of revisions and responses.

Point 1: For fig 1B, the authors state that the western blot indicates that the rPfHep protein has high specificity. A western blot shows whether the antibody is specific to a protein, not the other way. Why do they see two bands for the GST control in lane 2?

Response 1: Thank you for your constructive suggestion.

In the part of introduction, we added some message related to iron cycle. We repeated the SDS-PAGE and Western blot experiments, with lane 1 and lane 2 being rPfHep protein and empty vector (pGEX-KG)-expressed GST-tagged protein, respectively. It was indicated by the final results that the GST control in lane 2 was a single band. In this experiment, we employed new and highly specific antibody for the purpose of arriving at a reliable conclusion. In the meanwhile, we replaced relevant results in the manuscript (Fig.1 A, Fig.1 B). With regard to the anti-GST tag mouse monoclonal antibody (1: 4,000; 1 mg/mL) (Beyotime, Shanghai) and the HRP Goat Anti-Mouse secondary antibody (1: 5,000; 1 mg/mL) (ab6789, Abcam), Western blot was adopted.

Moreover, we added a detailed process of Western blot experiment named “4.5 Western Blot Analysis” to the “Materials and Methods” in the manuscript.

Figure 1:

Point 2: From fig 3A, the authors state that there was a significant decrease in hepcidin mRNA with 17-B and FSL. The fold change is minimal. I wouldn't consider that a significant change.

Response 2: Thank you for your constructive suggestion and correction. 

We applied the T-test method to the analysis on the significance of control group, 17-β group and FSL group respectively, and finally came to the conclusion that the P value of FSL group was 0.028, which was significant, while that of the 17-β group was not. Furthermore, we examined all the results and revised Figure3A. I’d like to extend my apology to you for the error in data analysis.

Figure 3:

Point 3: I didn't understand what the first panel in Fig 3D is. I see no changes in any of those spots.

Response 3: Thank you for your professional comments.

In terms of Fig. 3D panel, a direct counting of Aeromonas veronii in Brain Heart Infusion agar medium, the colors of the colony and the medium are similar to each other, and adjustments to the parameters were not made during shooting, so the spots cannot be clearly seen. For fear of ambiguity, we removed the original Fig.3D in the manuscript, deleted the first pane of Fig. 3D, and displayed it separately in the supplement figure.

Figure 3:

S-Figure:

Point 4: In the experiments described in figure 4, the authors introduce a purified hepcidin through the gavage. Can they show the protein is stable under these conditions, and that it reaches the hepatocytes in an active form? If non, the authors can not make any conclusions from these data.

Response 4: I’d like to extend my gratitude to you for your professional comments and constructive suggestions.

Our experimental principle is as follows: hepcidin triggers iron efflux protein Ferroportin (Fpn) polyubiquitination and induces its internalization to regulate iron changes. Fpn exists in enterocytes, splenic macrophages, and Kupffer cells, as well as hepatocytes. Hepcidin incubated HEK293-Fpn cells for 4 hours and thus it resulted in Fpn degradation. Therefore, the effect imposed by hepcidin on cells was very extensive, which could be significant in a short period of time after gavage in Pelteobagrus fulvidraco.

It has been proved by our laboratory that Fpn degrades in HEK293T cells through the action of Ctenopharyngodon idella hepcidin. Phan-Aram labeled the Oreochromis niloticus Linn hepcidin[1] (sMat On-Hep) using FITC fluorescence, and tested it in the lymphocytes of peripheral blood mononuclear cells and tilapia, which consequently proved that On-Hep could play its role of iron regulation and the function of immune regulator[2].

Additionally, it has been proved by the studies in our laboratory and other laboratories that feeding hepcidin protein to fish can enable them to perform normal functions[3, 4].

  1. Hu, Y.; Kurobe, T.; Liu, X.; Zhang, Y.A.; Su, J.; Yuan, G. Hamp Type-1 Promotes Antimicrobial Defense via Direct Microbial Killing and Regulating Iron Metabolism in Grass Carp (Ctenopharyngodon idella). Biomolecules 2020, 10, doi:10.3390/biom10060825.
  2. Phan-Aram, P.; Mahasri, G.; Kayansamruaj, P.; Amparyup, P.; Srisapoome, P. Immune Regulation, but Not Antibacterial Activity, Is a Crucial Function of Hepcidins in Resistance against Pathogenic Bacteria in Nile Tilapia (Oreochromis niloticus Linn.). Biomolecules 2020, 10, doi:10.3390/biom10081132.
  3. Chen, T.; Zhou, J.; Qu, Z.; Zou, Q.; Liu, X.; Su, J.; Fu, X.; Yuan, G. Administration of dietary recombinant hepcidin on grass carp (Ctenopharyngodon idella) against Flavobacterium columnareinfection under cage aquaculture conditions. Fish Shellfish Immunol 2020, 99, 27-34, doi:doi:10.1016/j.fsi.2020.01.042.
  4. Ting, C.-H.; Pan, C.-Y.; Chen, Y.-C.; Lin, Y.-C.; Chen, T.-Y.; Rajanbabu, V.; Chen, J.-Y. Impact of Tilapia hepcidin 2-3 dietary supplementation on the gut microbiota profile and immunomodulation in the grouper (Epinephelus lanceolatus). Scientific Reports 2019, 9, doi:10.1038/s41598-019-55509-9.

Reviewer 2 Report

The authors describe the expression and purified recombinant yellow catfish (Pelteobagrus fulvidraco) hepcidin 19 protein. They, in addition, provide evidence that the recombinant protein promotes liver iron accumulation as a strategy to limit pathogen growth due to iron starvation. The work has implications for how bacterial virulence is mitigated under iron limitation conditions.

I have a few comments for the authors.

  1. The figures showing a gel and western blot have extremely low resolution, something even below what any smartphones these days would be able to capture. Please, provide better-quality images.
  2. The figure legends are poorly described; please provide more information and a brief description of the staining of histopathological preparation. It is normal to have iron accumulation in clusters or it should be more homogenous within the organ, please explain. Add to the description also the statistical tests. In figure 4A, is there statistics (Kaplan-Meir?), please highlight.
  3. The introduction is quite short. It would benefit from a more extensive review of the state of the art, and from contextualizing iron limitation as a strategy across different species (i.e discuss other genes such as transferrin, ferritin, lactoferrin, and others).   
  4. Figure 3D, the colony counts are unnecessary, particularly if the colonies are almost impossible, please, remove or add it to a SM figure with the proper size.
  5. The author used beta-actin as a housekeeping gene, did this gene was validated to ensure that in this condition its expression remains no significantly changed?
  6. In the method of bacterial lysate preparation, bacteria were collected by centrifugation at 5000 × g for 10 min, 377 re-suspended in 20 mM PBS (pH 6.3), and high-pressure crushed on ice… what method is this? French press? Please, clarify.
  7. This study clearly has implications for aquaculture. Does the accumulation of iron in the liver limit the fitness of catfishes or impact on productivity?

Author Response

We appreciate your constructive comments and suggestions. We apologize for submitting an inadequately well-presented manuscript, thus causing difficulty in understanding the presented material and wasting your time. Following your suggestion, we have carefully proofread and reorganized the manuscript, improved wherever we found room for improvement.

Point 1: The figures showing a gel and western blot have extremely low resolution, something even below what any smartphones these days would be able to capture. Please, provide better-quality images.

Response 1: Thank you for your professional comments and constructive suggestion. 

We conducted the SDS-PAGE and Western blot experiments again, replaced new antibodies and adopted better shooting equipment to get gel and Western blot images of higher quality. In the end, we replaced the original Fig. 1A and Fig. 1B with these images.

Figure 1:

Point 2: The figure legends are poorly described; please provide more information and a brief description of the staining of histopathological preparation. It is normal to have iron accumulation in clusters or it should be more homogenous within the organ, please explain. Add to the description also the statistical tests. In figure 4A, is there statistics (Kaplan-Meir?), please highlight.

Response 2: I’d like to extend my gratitude to you for your professional comments and constructive suggestions.

(1)Firstly, we added a detailed process of Hematoxylin and eosin (HE) staining, prussian blue staining experiment named “4.8 Tissue staining” to the “Materials and Methods” in the manuscript.

“The livers and gills were dissected and fixed immediately in 10% neutral buffered formalin for 24 h, dehydrated, paraffin-embedded, and sectioned. Sectioned samples (4 m) were mounted on aminopropyl-triethoxysilane-coated slides. Following the deparaffinization in xylene, sections were rehydrated, stained with Hematoxylin and eosin (HE) and prussian blue staining, and mounted with neutral gum, in which the images were captured.”

(2)Secondly, in the legends of Figure 5, we added a statement to describe the degree of lesion in each group.

“In the control group, the proliferation of gill filament epithelial cells was observed, and the gill pieces were severely proliferated and fused into a plate shape. As a result, the histopathological changes of rPfHep group and BBR group were better than that of the control group.” “In the control group, hepatocytes were enlarged; inflammatory cell infiltration and even focal necrosis appeared in the liver. However, it was not so serious in the rPfHep group and BBR group.”

(3)Thirdly, no iron deposition was available in normal liver tissue, and it was rare to see blue granules in prussian blue staining. When the body was inflamed, iron accumulated in the organs and formed larger particles.

(4)Fourthly, we added the statistical tests to the manuscript.

“We applied Kaplan-Meier test to the analysis on mortality among groups. It was identified that the mortality rate in the rPfHEP group and BBR group was significantly lower than that of the control group (P<0.01).”

Point 3: The introduction is quite short. It would benefit from a more extensive review of the state of the art, and from contextualizing iron limitation as a strategy across different species (i.e discuss other genes such as transferrin, ferritin, lactoferrin, and others).

Response 3: Thank you for your constructive suggestion.We added some message related to iron limitation to the introduction.

“There is a close relationship between transferrin and the immune system. Transferrin binds to iron, creating an environment with iron of low levels, where few microorganisms can survive and prosper[22, 23]. Transferrin exists in serum, and lactoferrin is secreted by the mucosa, as well as neutrophils at infection sites, and it serves as a host iron-withholding response, sequestering iron away from invading microorganisms. Apart from that, all cells contain ferritin, a protein composed of 24 subunits. Concerning ferritin, it can bind up to 4500 iron atoms, allowing it to be the major protein responsible for iron storage in cells[24]. Ferritin-bound iron, the major mechanism of iron storage in macrophages and liver hepatic cells[25], sequesters iron upon the increase of its intracellular levels, detoxifying and avoiding damage. Liver ferritin contains 50% of iron corporal reserves[26].”

Point 4: Figure 3D, the colony counts are unnecessary, particularly if the colonies are almost impossible, please, remove or add it to a SM figure with the proper size.

Response 4: Thank you for your professional comments.

In terms of Fig. 3D panel, a direct counting of Aeromonas veronii in Brain Heart Infusion agar medium, the colors of the colony and the medium are similar to each other, and adjustments to the parameters were not made during shooting, so the spots cannot be clearly seen. For fear of ambiguity, we removed the original Fig.3D in the manuscript, deleted the first pane of Fig. 3D, and displayed it separately in the supplement figure.

Figure 3:

S-Figure:

Point 5: The author used beta-actin as a housekeeping gene, did this gene was validated to ensure that in this condition its expression remains no significantly changed?

Response 5: Thank you for your professional comments.

It has been demonstrated by our laboratory that the expression of beta-actin gene in Pelteobagrus fulvidraco is stable, which was verified by Zhu and Chen. In addition, no existing studies suggest that the beta-actin and hepcidin genes affect each other’s expression. Hence, I directly employed beta-actin as the housekeeping gene in the experiment.

Zhu, W.; Zhang, Y.; Zhang, J.; Yuan, G.; Liu, X.; Ai, T.; Su, J. Astragalus polysaccharides, chitosan and poly(I:C) obviously enhance inactivated Edwardsiella ictaluri vaccine potency in yellow catfish Pelteobagrus fulvidraco. Fish & Shellfish Immunology 2019, 87, 379-385, doi:10.1016/j.fsi.2019.01.033.

Chen, H.J.; Yuan, G.L.; Su, J.G.; Liu, X.L. Hematological and immune genes responses in yellow catfish (Pelteobagrus fulvidraco) with septicemia induced by Edwardsiella ictaluri. Fish & Shellfish Immunology 2020, 97, 531-539, doi:10.1016/j.fsi.2019.11.071.

Point 6: In the method of bacterial lysate preparation, bacteria were collected by centrifugation at 5000 × g for 10 min, 377 re-suspended in 20 mM PBS (pH 6.3), and high-pressure crushed on ice… what method is this? French press? Please, clarify.

Response 6: Thank you for your professional comments.

We employed low temperature induction and high pressure cleavage respectively with the aim of promoting the expression of soluble protein rPfHep and breaking bacterial cells to facilitate the release of recombinant PfHep protein. Subsequently, the supernatant containing the protein was collected, and then the purified protein was obtained as well through affinity chromatography.

In the entire process, we referred to the experimental methods previously adopted by Xiao and Chen.

Xiao, X.; Zhang, Y.Q.; Liao, Z.W.; Su, J.G. Characterization and Antimicrobial Activity of the Teleost Chemokine CXCL20b. Antibiotics-Basel 2020, 9, 13, doi:10.3390/antibiotics9020078.

Chen, T.; Zhou, J.; Qu, Z.; Zou, Q.; Liu, X.; Su, J.; Fu, X.; Yuan, G. Administration of dietary recombinant hepcidin on grass carp (Ctenopharyngodon idella) against Flavobacterium columnare infection under cage aquaculture conditions. Fish Shellfish Immunol 2020, 99, 27-34, doi:doi:10.1016/j.fsi.2020.01.042.

Point 7: This study clearly has implications for aquaculture. Does the accumulation of iron in the liver limit the fitness of catfishes or impact on productivity?

Response 7: Thank you for your professional comments.

Inflammation stimulated increased production of the iron-regulatory peptide, hepcidin, by hepatocytes and the pro-inflammatory cytokine, IL-6, which suppressed erythropoiesis. When inflammation resolved, the levels of hepcidin and IL-6 declined, allowing iron to be exported from macrophages to erythrocytes, promoting erythropoiesis, and stimulating iron releasing from the liver and returning to the internal circulation[1].

Ting conducted an experiment with 28-day diet supplemented with hepcidin in Epinephelus lanceolatus[2], and our laboratory conducted an experiment with14-day diet supplemented with hepcidin in Ctenopharyngodon idella[3]. It was shown by the final results that long-term addition of hepcidin failed to influence the growth of fish but it indeed improved the immunity of Pelteobagrus fulvidraco.

It is easy to come to the conclusion that iron accumulation in the liver functions as a protective measure of the body against infection and it will return to the normal state after the disappearance of inflammation, rather than affecting the growth of Pelteobagrus fulvidraco.

  1. Ganz, T. Iron and infection. International Journal of Hematology 2018, 107, 7-15, doi:doi:10.1007/s12185-017-2366-2.
  2. Ting, C.-H.; Pan, C.-Y.; Chen, Y.-C.; Lin, Y.-C.; Chen, T.-Y.; Rajanbabu, V.; Chen, J.-Y. Impact of Tilapia hepcidin 2-3 dietary supplementation on the gut microbiota profile and immunomodulation in the grouper (Epinephelus lanceolatus). Scientific Reports 2019, 9, doi:10.1038/s41598-019-55509-9.
  3. Chen, T.; Zhou, J.; Qu, Z.; Zou, Q.; Liu, X.; Su, J.; Fu, X.; Yuan, G. Administration of dietary recombinant hepcidin on grass carp (Ctenopharyngodon idella) against Flavobacterium columnareinfection under cage aquaculture conditions. Fish Shellfish Immunol 2020, 99, 27-34, doi:doi:10.1016/j.fsi.2020.01.042.

Round 2

Reviewer 1 Report

I am happy with the way the authors have addressed all my concerns.